# Adversarial Robust Aerial Image Recognition Based on Reactive-Proactive Defense Framework with Deep Ensembles

**Zihao Lu, Hao Sun \*, Kefeng Ji and Gangyao Kuang**

The State Key Laboratory of Complex Electromagnetic Environment Effects on Electronics and Information System, National University of Defense Technology, Changsha 410073, China; luzihao21@nudt.edu.cn (Z.L.); jikefeng@nudt.edu.cn (K.J.); kuanggangyao@nudt.edu.cn (G.K.)

\* Correspondence: sunhao@nudt.edu.cn

**Abstract:** As a safety-related application, visual systems based on deep neural networks (DNNs) in modern unmanned aerial vehicles (UAVs) show adversarial vulnerability when performing real-time inference. Recently, deep ensembles with various defensive strategies against adversarial samples have drawn much attention due to the increased diversity and reduced variance for their members. Aimed at the recognition task of remote sensing images (RSIs), this paper proposes to use a reactive-proactive ensemble defense framework to solve the security problem. In reactive defense, we fuse scoring functions of several classical detection algorithms with the hidden features and average output confidences from sub-models as a second fusion. In terms of proactive defense, we attempt two strategies, including enhancing the robustness of each sub-model and limiting the transferability among sub-models. In practical applications, the real-time RSIs are first input to the reactive defense part, which can detect and reject the adversarial RSIs. The accepted ones are then passed to robust recognition with a proactive defense. We conduct extensive experiments on three benchmark RSI datasets (i.e., UCM, AID, and FGSC-23). The experimental results show that the deep ensemble method of reactive and proactive defense performs very well in gradient-based attacks. The analysis of the applicable attack scenarios for each proactive ensemble defense is also helpful for this field. We also perform a case study with the whole framework in the black-box scenario, and the highest detection rate reaches 93.25%. Most of the adversarial RSIs can be rejected in advance or correctly recognized by the enhanced deep ensemble. This article is the first one to combine reactive and proactive defenses with a deep ensemble against adversarial attacks in the context of RSI recognition for DNN-based UAVs.

**Keywords:** deep neural network; adversarial defense; deep ensemble; unmanned aerial vehicle; remote sensing; image recognition

## 1. Introduction

In recent years, the prevalence of deep neural networks (DNNs) has tremendously boosted the automatic interpretation of high-resolution remote sensing images (RSIs) [1–4]. State-of-the-art performances of various tasks in the field of remote sensing have been achieved by advanced DNN models. Concurrently, an increasing number of modern unmanned aerial vehicles (UAVs) with DNN-based visual navigation and recognition systems are able now to support on-device inference for real-time RSIs and quickly provide useful analysis of the images for both military (e.g., target recognition [5–7], battlefield surveillance [8–10], communication [11]) and civilian (e.g., land planning [12], medical rescue [13], parcel delivery [14]) use.

Despite acquiring such great success, DNN models show serious vulnerability when confronting adversarial examples [15–17], which are part of an emerging wave of anti-UAV efforts. Once attackers obtain the original input, they can add well-designed imperceptible perturbations to the benign data with the aim of maliciously degrading the performance of models on UAVs and causing harmful effects.

For example, when a UAV performs target recognition and tracking tasks in a military mission, the adversary evades the recognition of valuable targets by making the UAV track another one, or disrupting the UAV's visual navigation to induce an emergency landing [18], a reality which creates greater demand for the accurate perception of the surrounding environment in the DNN-based visual system. As shown in Figure 1, we consider the case of an attacker illegally accessing the commutation approach (e.g., Wi-Fi) that transmits images between the UAV and controller and manipulating the real-time RSIs from sensors to cause mispredictions when performing a target recognition task. Adversarial vulnerability of DNN models is also an effective and seriously disruptive problem in object detection [19,20] and EEG signal processing [21] tasks. Moreover, previous research [22–25] has demonstrated that adversarial examples generated against a surrogate model are able to mislead target model with high probability due to the similar feature representations. This property of adversarial transferability can greatly reduce the difficulty of mounting an attack and raise the threat to DNN-based applications.

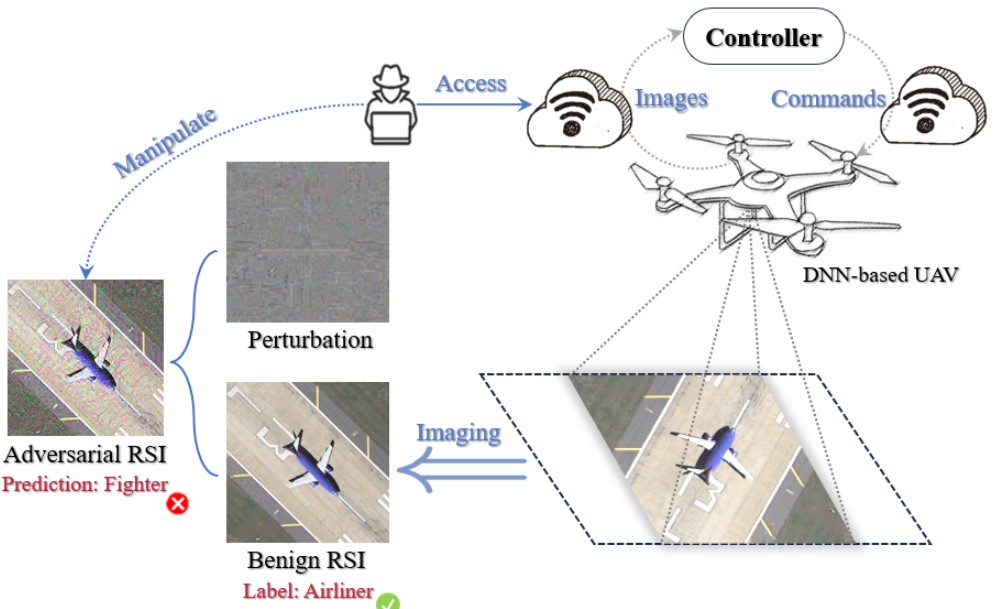

**Figure 1.** One possible scenario is that adversaries can craft adversarial RSIs using the communication approach, which causes misleading outputs of the DNN model on a modern UAV when performing a target recognition task.

Faced with the safety problem of DNN models, researchers have proposed plenty of adversarial defense methods, mainly in the context of natural images, which can be divided into two categories: proactive and reactive defenses. Proactive defense aims at robust classification of all the adversarial examples while maintaining high test accuracy in the absence of attacks [26–32]. However, generating a sufficiently robust model is impractical and sometimes the robust classifier's decision can disagree with the decision from a non-robust classifier [33]. Thus, reactive defense (i.e., adversarial detection) has been regarded as an alternative, which solves a binary classification problem of whether the input is attacked [34–38]. For both types of defenses, most strategies are concerned with a stand-alone model. Nevertheless, the stand-alone model still has the risk of being targeted again by stronger and unknown attacks after adversarial robustness enhancement. Thus, we consider using deep ensemble models in both proactive and reactive defense against the adversarial vulnerability in DNN-based UAVs.

Deep ensembles are intuitively expected to be more robust than single networks due to the increased diversity and reduced variance for their members in the proactive defense [39]. An attack against the ensemble can succeed only when most sub-models converge toward wrong predictions [40]. However, simply assembling DNNs and fusing

their outputs may not be very effective. There are mainly two ways to reinforce the deep ensembles against adversarial attacks: improving the robustness of each sub-model and reducing the adversarial transferability within the ensemble. Because deep ensembles are not fully explored in the remote sensing field, we cannot tell which of them performs better in an adversarial scenario.

Therefore, both approaches for the proactive defense of deep ensembles will be compared. On the one hand, we train each sub-model by adversarial training (AT). Although AT can force DNNs to learn robust features and thereby improve their robustness, the deactivation of non-robust ones discards some discriminative cues that are beneficial for correct model prediction. To recapture the useful information and given the simplicity for edge devices like UAVs, we attach an easy-to-plugin module, feature separation and recalibration (FSR) [41] module to an intermediate layer within each sub-model. On the other hand, inspired by field theory in physics research, adversarial transferability is revisited from the perspective of the loss field. By promoting the orthogonality between the loss fields and reducing the loss field strength of sub-models, we introduce two penalty terms that can limit the adversarial transferability into the original loss and optimize the ensemble.

Deep ensembles can also be applied in reactive defenses. One popular adversarial detection strategy is based on the scores of statistical metrics computed from the hidden features extracted by DNNs. Then, the scores are sent into a traditional classifier, such as logistic regression and support vector machine, to determine if the input is adversarial or not. Since different statistical metrics can capture distinct properties in distinguishing the adversarial examples from benign ones [42], it seems natural that combining scores from multiple detection algorithms can lead to a better performance in RSIs with rich spectral and spatial information. Furthermore, we can repeat the above process for diverse DNNs in an ensemble and average the confidence scores from the final traditional classifiers on the decision level as a second fusion.

In this article, we consider an intractable case where a DNN-based UAV with a visual recognition system suffers from various adversarial attacks. The objects to be recognized for the UAV include scenes and targets. To cope with the problem, we will provide a reactive-proactive defense framework based on deep ensembles. Specifically, the real-time RSIs are first input into a deep ensemble model of reactive defense. If the detection gives a negative result, the deep ensemble of proactive defense will be activated to perform a robust recognition for the malicious RSIs. We will compare two strategies for enhancing the adversarial robustness in the proactive defense, analyzing the applicable attack scenarios for each strategy. The ensemble methods will also be compared with stand-alone models for both defenses.

Extensive experiments are conducted on three optical datasets with RSIs captured by UAVs, which are UC Merced land-use (UCM) [43], Aerial Image Dataset (AID) [44], and Fine-grained Ship Collection-23 (FGSC-23) [45]. As a lightweight and commonly-used architecture for edge devices, ResNet-18 [46] network architecture with randomly initialized parameters serves as the sub-model in a deep ensemble. The experimental results show that the proactive-reactive defense framework with deep ensembles is effective in enhancing the robustness of intelligent recognition systems on UAVs from malicious RSIs. The main contributions and findings of our work are listed as follows.

- We combine the different scoring functions of classic adversarial detection algorithms on the DNN feature level and average the outputs of sub-models on the decision level in the reactive defense.
- We develop a novel regularized way to train the deep ensemble based on promoting the orthogonality of the sub-model's loss field and reducing the loss field strength of sub-models.
- We investigate two ways of proactively enhancing the adversarial robustness (i.e., enhancing each sub-model and limiting adversarial transferability among sub-models) of deep ensembles and analyze their suitable attack scenarios based on the experimental results.

- For the first time, we combine the reactive defense and proactive defense with deep ensembles in sequence to resist adversarial attacks for the RSI recognition task. The defensive structure is called reactive-proactive defense framework with deep ensembles, which can discard the data or start enhanced deep ensembles when the input RSI is detected as adversarial, for example, in order to protect the target model during performing important tasks.
- We perform a case study of the whole reactive-proactive framework in a black-box scenario. The system can reject most adversarial RSIs in the reactive defense part and perform robust recognition for the remaining RSIs.

The rest of this paper is organized as follows. Section 2 introduces the related works in this article. Section 3 introduces the deep ensemble method used in reactive defense, two strategies in proactive defense, and the reactive-proactive ensemble defense framework in detail. Section 4 reports on the experimental results and provides an analysis. Finally, conclusions are given in Section 5.

## 2. Related Works

### 2.1. Adversarial Attacks

Since the work of Szegedy et al. [22] demonstrated the existence of adversarial examples, ensuring robustness against adversarial attacks has been crucial to modern DNN systems. Given an original RSI $x \in \mathbb{R}^{h \times w \times c}$ with a true label $y \in \mathcal{Y} = \{0, 1, ..., C\}$, an adversarial attack usually adds small noise $\delta \in \mathbb{R}^{h \times w \times c}$ to form the adversarial example $x_{adv} = x + \delta$ where $h$, $w$ and $c$, respectively, represent the height, width, and number of channels of the RSI. Attackers can maximize the loss function $\mathcal{L}_\theta$ of training the DNN model $f$ with the trained parameters $\theta$. The objective of attackers can be formulated as (1).

$$x_{adv} = x + \arg \max_{\delta \in \mathcal{A}} \mathcal{L}_\theta(x + \delta, y) \tag{1}$$

In (1), $\mathcal{A}$ is a constraint to ensure $x_{adv}$ visually indistinguishable from $x$. The constraint $\mathcal{A}$ typically has the form $\|\delta\|_p < \epsilon$ with perturbation scale $\epsilon$ and $l_p$-norm. Furthermore, various adversarial attack algorithms have been proposed to guide attackers to modify the pixel values, such as the fast gradient sign method (FGSM) [26], basic iterative method (BIM) [47], projected gradient descent (PGD) [48], Carlini and Wanger attack (C&W) [49], momentum iterative method (MIM) [50], and Deepfool [51], which are commonly considered in the studies of adversarial defense.

During the performance of the visual recognition tasks by DNN-based UAVs, we assume that adversaries can spoof the messages and manipulate the real-time RSIs from sensors by illegally accessing the communication approach (e.g., Wi-Fi network) that transmits images among UAVs [52]. In this article, we mainly focus on the $l_\infty$-bounded white-box threat model, which grants attackers complete access to DNN models. In addition, we also provide a case study for black-box transfer attacks. Black-box scenarios assume that the adversaries generate adversarial RSIs with a surrogate model and transfer them to fool the targeted model.

### 2.2. Adversarial Vulnerability in DNN-Based UAVs

Previous papers have indicated the adversarial vulnerability in DNN models for RSIs, leading to a security problem in modern UAVs. They mainly involve scene classification [53–55] and target recognition [56–58] for digital attacks, which are consistent with the supposed threat model in this article. Moreover, there are some explorations into adversarial patches that can be printed and applied to a physical scene or target [59–61]. Direct discussions about the vulnerabilities of UAVs also exist, and some of them give an analysis from the viewpoint of adversarial attacks [18,62,63]. However, few of them perform an effective defense with experiments, and we provide the ensemble strategies to address the problem.

### 2.3. Adversarial Ensemble Defense

Ensemble learning has been extensively studied to improve performance and achieve better generalization. Several individual sub-models in an ensemble can produce weak results and fuse their information to obtain a better final decision via adaptive voting mechanisms [64,65]. With the early success in basic images classification task with bagging [66] and AdaBoost [67], Kuncheva et al. [68] evaluated the relationship between ensemble accuracy and sub-model diversity. They found that higher diversity within the ensemble can be beneficial in obtaining better results. In addition, Zhang et al. [69] utilized pairwise constraints among sub-models to improve diversity. Li et al. [70] found that the diversity of the ensemble could be utilized as regularization for better generalization.

Recently, constructing deep ensemble models has been an intriguing direction in defending DNN models against adversarial attacks. The initial success was also from the fusion of outputs from multiple DNNs [71]. However, the adversarial transferability property among sub-models greatly limits further enhancements with adversarial examples for deep ensembles. Thus, an intuition is that we can mitigate adversarial transferability by promoting a form of diversity within the ensemble, and a recent line of studies has verified its effectiveness. Specifically, Pang et al. [72] proposed ADP, which guides sub-models to output diverse non-maximal predictions. Sen et al. [73] proposed EMPIR to enforce ensemble diversity by extreme model quantization. Yang et al. [40] proposed DVERGE, which isolates and diversifies the adversarial vulnerability by distilling non-robust features to induce diverse outputs. Kariyappa et al. [74] proposed GAL, which maximizes the gradient's orthogonality to reduce the overlap between sub-models' adversarial sub-spaces. Yang et al. [75] trained a robust ensemble by promoting input gradient diversity and model smoothness between base models.

The mentioned studies are based on proactive defense, and there is also some research in reactive defense with ensemble methods [42,76–78]. Most of them integrated several adversarial detection algorithms as an ensemble. Based on their strategies, we repeat the process on multiple DNNs and fuse the output confidence scores on the decision level. However, no articles have explored deep ensembles with RSIs under adversarial attack, and we fill the vacancy with a reactive-proactive defense framework in this article.

## 3. Methodology

In this section, we will introduce the deep ensemble models used in both reactive and proactive defense to address the security problem in DNN-based UAVs. Next, we will connect the two parts to form a system and introduce the metric for analyzing the defensive effect of this complete framework.

### 3.1. Reactive Defense

The reactive defense manifests its value when the output of the baseline model disagrees with that of a robust classifier, and we want to know if it is because the input has been attacked. Thus, in this article, we consider the adversarial detection part as a third party that can recommend if the input is benign or adversarial, which can be filtered out in advance [33].

We adopt two types of fusions with the idea of ensemble in the reactive defense part. Given the complex information and rich features in most RSIs, the first integration is combining the scores of different statistical metrics for the hidden features extracted by DNNs [42]. Specifically, for a real-time RSI $x_0$, the members in the ensemble of scoring functions are $d_1(a_l(x_0)), d_2(a_l(x_0)), ..., d_n(a_l(x_0))$ with the activation $a_l(x_0)$ at layer $l$ of DNN model $f(x_0), l \in 1, 2, ..., L$. The score vector of detector $i$ can be denoted as $\mathbf{D_i(x_0)}$ and written as (2).

$$\mathbf{D_i(x_0)} = \{d_i(a_1), d_i(a_2), ..., d_i(a_L)\}, i = 1, 2, ..., n \tag{2}$$

All the *n* score vectors compose a new larger vector $\mathbf{V}(\mathbf{x_0})$, which is expressed as (3).

$$\mathbf{V}(\mathbf{x_0}) = \{\mathbf{D_1}(\mathbf{x_0}), \mathbf{D_2}(\mathbf{x_0}), ..., \mathbf{D_n}(\mathbf{x_0})\} \tag{3}$$

Then, we train a simple logistic regression to compute the posterior probability, namely the confidence score of judging the input RSI as an attacked one (adv) as (4).

$$p(adv \mid \mathbf{V}(\mathbf{x_0})) = \left(1 + \exp\left(\beta_0 + \boldsymbol{\beta}^T \cdot \mathbf{V}(\mathbf{x_0})\right)\right)^{-1} \tag{4}$$

In (4), $\boldsymbol{\beta}^T$ is the weight vector for fitting the training data and adjusting the importance of the n detectors. We label the adversarial RSIs generated with PGD as the positive class and label the benign RSIs as the negative class.

The second fusion is performed on the decision level. We define an ensemble of DNN as $\mathcal{F} = \{f_1, f_2, \ldots, f_N\}$ and repeat the above process on *N* sub-models in the deep ensemble. The final decision can be obtained by averaging their output confidence scores $p_1, p_2, ..., p_N$ as (5).

$$p = \frac{1}{N} \cdot \sum_{i=1}^{N} p_i(adv \mid \mathbf{V}(\mathbf{x_0})) \tag{5}$$

The procedures can be illustrated in detail as Figure 2. If *p* is greater than 0.5, the ensemble model of reactive defense will determine the real-time RSI as an adversarial one, and benign otherwise.

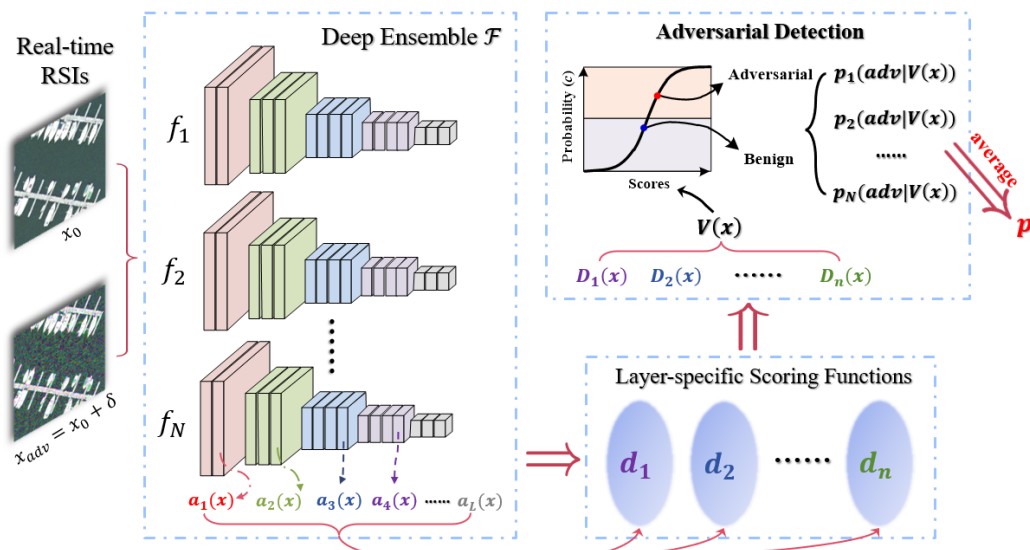

**Figure 2.** The illustration of the deep ensemble method used in reactive defense.

When training the detection algorithms to obtain proper thresholds and better generalization, we craft noisy examples with Gaussian noise as the benign examples. For the first integration, several statistical metrics can effectively mitigate the problems of potential overfitting and poor generalization when using only one detector. The final determination of adversarial RSIs can also benefit from the second integration for reducing the variance of feature representations in the DNN sub-models. Thus, we achieve the fusions on the feature and decision level simultaneously in this part.

The evaluation criteria in reactive defense include the area under the receiver operating characteristic (AUROC), the area under precision recall (AUPR), and the detection rate (DR). The correct detection of adversarial and benign RSIs are treated as true positives (TP) and true negatives (TN); conversely, the wrong detection of adversarial and benign RSIs

are false negatives (FN) and false positives (FP), respectively. *DR* is the proportion of *TP* and *TN* in all RSIs that participated in the test as (6).

$$DR = \frac{TP + TN}{TP + TN + FN + FP} \tag{6}$$

The reactive defense will let *TN* and *FN* pass and reject *TP* and *FP*. Thus, we need to increase the proportion of *TP* and *TN* for better detection performance.

### 3.2. Proactive Defense

In proactive defense, we aim to correctly recognize all the RSIs, including the benign RSIs and adversarial RSIs with deep ensemble models. To enhance the adversarial robustness of deep ensembles while maintaining high benign accuracy, we attempt the following two approaches.

### 3.2.1. Sub-Model Robustness Enhancement

One of the most intuitive ideas of adversarial robustness enhancement for deep ensembles is to improve the robustness of each member. Among the various empirical strategies for proactively reinforcing the individual DNNs, AT has come to prominence for its reliability and effectiveness. We formulate AT as a min-max optimization that minimizes the training loss with online-generated adversarial examples. The objective of AT can be expressed as (7).

$$\min_{\theta} \mathbb{E}_{(x,y) \sim \mathcal{D}} \left[ \max_{\delta \in \mathcal{A}} \mathcal{L}_{f_\theta}(x + \delta, y) \right] \tag{7}$$

In (7), the inner maximization focuses on the generation of adversarial examples with the underlying data distribution $\mathcal{D}$, which are often performed by PGD for greater robustness. The outer minimization optimizes parameters in the DNN to learn the recognition of adversarial examples with true labels. PGD-AT encourages the model to capture robust features and neglects some non-robust features. Non-robust features are those that cannot constitute the generic patterns for model correct predictions and can be targeted by attackers. However, the non-robust activations still contain plenty of discriminative cues that are not exploited for enhancing feature robustness.

In terms of the RSIs with rich features and complex ground objects, the mentioned phenomena are significantly obvious. We compute the frequency of channel-wise activations on UCM and AID datasets. As shown in Figure 3, adversarial RSIs activate the intermediate channels more uniformly in standard supervised training, especially for those that are less activated by benign RSIs (i.e., the right regions of blue columns). The low-activated channels for benign RSIs correspond to the non-robust activations, and adversarial RSIs tend to excite them frequently to cause variations and increase vulnerability. In Figure 4, the choice of activated channels is similar in the DNNs trained by PGD-AT on both RSI datasets. PGD-AT can force the models to deactivate the non-robust parts and make the adversarial RSIs behave like the benign ones.

To re-activate the information in non-robust features during AT, given the simplicity and quick inference on the edge devices of UAVs, we introduce a novel module FSR [41] into each sub-model of the deep ensemble like a plugin and still train in an end-to-end manner with little extra computational overhead. The FSR module is attached at an intermediate layer, including a separation and re-calibration stage.

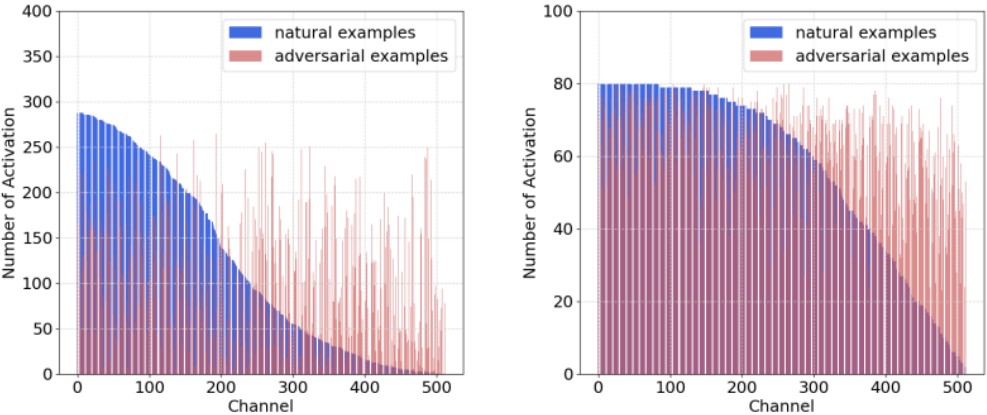

**Figure 3.** The activation frequency at layer 4 of ResNet-18 (512 channels) trained with standard supervised training (**Left**: AID dataset; **Right**: UCM dataset). Blue columns represent benign RSIs, and the red ones represent the PGD-20 adversarial RSIs.

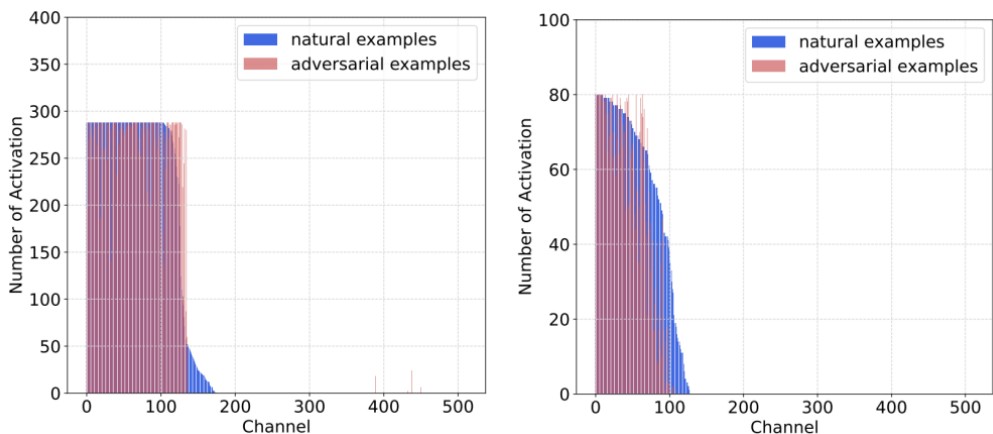

**Figure 4.** The activation frequency at layer 4 of ResNet-18 (512 channels) trained with AT (**Left**: AID dataset; **Right**: UCM dataset). Blue columns represent benign RSIs, and the red ones represent the PGD-20 adversarial RSIs.

During the separation stage, we define a separation net that can learn the robustness of each feature unit and output a robustness map $r \in \mathbb{R}^{h \times w \times c}$, where $h$, $w$ and $c$ are the height, width, and number of channels of a selected feature map $\gamma \in \mathbb{R}^{h \times w \times c}$. The robustness map contains scores for each unit, and a higher score corresponds to a more robust activation. To disentangle the feature map, we define a soft mask $s \in \mathbb{R}^{h \times w \times c}$ based on $r$ with Gumbel softmax [79] as (8).

$$s = \frac{e^{((\log(\sigma(r)) + g_1)/\tau)}}{e^{((\log(\sigma(r)) + g_1)/\tau)} + e^{((\log(1-\sigma(r)) + g_2)/\tau)}} \tag{8}$$

where $\sigma(\cdot)$ is the sigmoid function and $g_1$ and $g_2$ represent the random values sampled from the Gumbel distribution. $\tau$ is a temperature to control $g_1$ and $g_2$. We can set a threshold value, and the scores larger than it correspond to the robust features, which are denoted as $\gamma^+$. The remaining are non-robust features $\gamma^-$. To guide the learning of robustness scores based on their influences on making predictions for each sub-model, an additional MLP layer is attached after the separation stage and input into $\gamma^+$ and $\gamma^-$. Specifically, we formulate the guidance process with a loss function $\mathcal{L}_1$ as (9).

$$\mathcal{L}_1 = -\sum_{c=1}^{C} \left( y_c \cdot \log(p_c^+) + y_c' \cdot \log(p_c^-) \right) \tag{9}$$

In (9), $y_c$ is the true label and $p_c^+$ corresponds to its confidence score. $p_c^-$ is the highest score among the wrong labels $y_c'$. $\mathcal{L}_1$ guides the FSR module to assign a high robustness score to the feature units that can predict correctly as the robust features.

After obtaining $\gamma^+$ and $\gamma^-$, we introduce the re-calibration net $\mathcal{R}(\cdot)$ to adjust the non-robust part $\gamma^-$ to $\tilde{\gamma}^-$ as (10).

$$\tilde{\gamma}^- = \gamma^- + m^- * R(\gamma^-) \tag{10}$$

We also attach an additional MLP-based layer based on $\tilde{\gamma}^-$ to guide the adjustment of $\gamma^-$ to make it useful for model correct predictions, which can also be expressed as a loss function $\mathcal{L}_2$ as (11):

$$\mathcal{L}_2 = -\sum_{c=1}^{C} y_c \cdot \log(\tilde{p}_c^-) \tag{11}$$

where $\tilde{p}_c^-$ is the confidence score for each label with $\tilde{\gamma}^-$ as the input to the additional layer. We then can obtain the new feature map $\tilde{\gamma} = \gamma^+ + \tilde{\gamma}^-$ with more robust feature units.

The overall process can be optimized during the training time and summarized as $\mathcal{L}_{FSR}$ in a compact way.

$$\mathcal{L}_{FSR} = \frac{1}{N} \cdot \sum_{i=1}^{N} \mathcal{L}_{adv}(f_i(x), y) + \lambda_{sep} \cdot \mathcal{L}_{1,i} + \lambda_{rec} \cdot \mathcal{L}_{2,i} \tag{12}$$

where $\lambda_{sep}$ and $\lambda_{rec}$ are hyperparameters that control the weights of $\mathcal{L}_{1,i}$ and $\mathcal{L}_{2,i}$ for sub-model $f_i(x)$. $\mathcal{L}_{adv}$ can be the loss of one variant of AT for training the deep ensemble.

We can illustrate the optimization process for the first strategy of proactively enhancing the adversarial robustness of deep ensembles, as shown in Figure 5, with a better understanding.

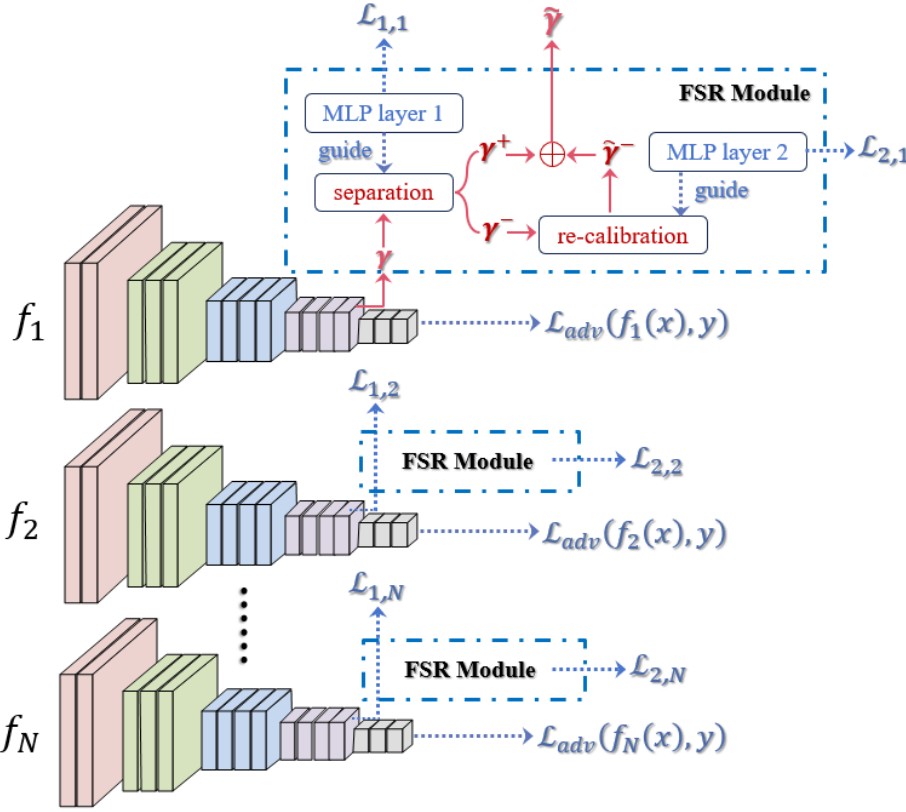

**Figure 5.** The illustration of optimization for the first strategy of proactively enhancing adversarial robustness of deep ensembles.

### 3.2.2. Adversarial Transferability Reduction

Recent studies [53,55] have shown that optical RSIs also suffer the adversarial transferability among DNN models; thus, the second attempt of proactively enhancing the robustness for deep ensembles is to reduce the transferability among the sub-models. We first quantify and characterize the property from a new perspective of loss field.

Given an RSI $x \in \mathbb{R}^{h \times w \times c}$, we can regard it as a point in $h \times w \times c$ dimensional data space. For one of its adversarial versions $x + \delta$, its adversarial risk is defined as $\eta'_{\mathcal{F}} = \Pr(\mathcal{F}(x + \delta) \neq y)$ and its adversarial empirical risk is defined as $\xi'_{\mathcal{F}}(x + \delta) = \mathbb{E}[\mathcal{L}_{\mathcal{F}}(x + \delta, y)]$. Generally speaking, $\eta'_{\mathcal{F}}$ is statistically proportional to $\xi'_{\mathcal{F}}$. Their relationship can be described as follows: $\eta'_{\mathcal{F}} = \Psi(\xi'_{\mathcal{F}})$, where $\nabla_{\xi'_{\mathcal{F}}} \eta'_{\mathcal{F}} > 0$. There is a unique loss corresponding to every point $x$, which forms the gradient field of loss with respect to the input space, i.e., loss field. We represent the loss field in the neighborhood space around the input RSI $x$ by the "electric field line" across $x$, which is the line with the largest field strength called the loss field line. From the perspective of the loss field, we can find the increased loss in a transfer attack is approximated as the integral of the target model's loss field over the surrogate model's loss field line within a certain range. Then we can quantify the transferability $\mathcal{T}$ as (13) between a surrogate model $f_1 \in \mathcal{F}$ and a target model $f_2 \in \mathcal{F}$ if the RSI is attacked against $f_1$ as $x_t = x + \delta_{f_1}$ and $\mathcal{F}$ is still a deep ensemble.

$$\mathcal{T} = \Pr(f_1(x_t) \neq y \wedge f_2(x_t) \neq y \mid f_1(x) = y \wedge f_2(x) = y) \tag{13}$$

From (13), we set a precondition that both the surrogate model and target model obtain correct results on the benign RSI $x$. Then, the adversarial transferability $\mathcal{T}$ is described as the probability that the adversarial RSI $x_t$ can fool $f_1$ and $f_2$ at the same time. Mathematically, we can further obtain the upper bound of (14):

$$\mathcal{T} \leq \frac{\eta_{f_1, f_2}}{1 - \eta_{f_1} - \eta_{f_2}} \tag{14}$$

In (14), $\eta_{f_1, f_2} = \Pr(f_2(x + \delta_{f_1}) \neq y)$, and an extended form of (14) is calculated as follows.

$$\mathcal{T} \leq \frac{\Psi\left(\xi_{f_2}\left(x + \delta_{f_1}\right)\right)}{1 - \eta_{f_1} - \eta_{f_2}}$$

$$\leq \frac{\Psi\left(\mathcal{L}_{f_2}(x, y) + \int_0^\epsilon \frac{\nabla_{\tilde{x}} \mathcal{L}_{f_2}(\tilde{x}, y) \cdot \nabla_{\tilde{x}} \mathcal{L}_{f_1}(\tilde{x}, y)}{\left\|\nabla_{\tilde{x}} \mathcal{L}_{f_1}(\tilde{x}, y)\right\|_2} \right) d(\tilde{x} - x)\right)}{1 - \eta_{f_1} - \eta_{f_2}} \tag{15}$$

In (15), $\epsilon$ is the perturbation scale (i.e., attack strength) and $\tilde{x} = x + x_1$ where $x_1$ is one point on the surrogate model's loss field line. Similarly, if we take another point $x_2$ on the target model's loss field line and define $\tilde{x}'$, then the second term in $\Psi(\cdot)$ is lower than (16).

$$\int_0^\epsilon \frac{\nabla_{\tilde{x}} \mathcal{L}_{f_2}(\tilde{x}, y) \cdot \nabla_{\tilde{x}} \mathcal{L}_{f_1}(\tilde{x}, y)}{\left\|\nabla_{\tilde{x}} \mathcal{L}_{f_1}(\tilde{x}, y)\right\|_2}\right) d(\tilde{x} - x)\right) \leq \frac{\beta}{\epsilon} \cdot$$

$$\int_0^\epsilon \int_0^\epsilon \left\|\nabla_{\tilde{x}} \mathcal{L}_{f_1}(\tilde{x}, y)\right\|_2 \frac{\nabla_{\tilde{x}} \mathcal{L}_{f_1}(\tilde{x}, y) \cdot \nabla_{\tilde{x}'} \mathcal{L}_{f_2}(\tilde{x}', y)}{\left\|\nabla_{\tilde{x}} \mathcal{L}_{f_1}(\tilde{x}, y)\right\|_2 \left\|\nabla_{\tilde{x}'} \mathcal{L}_{f_2}(\tilde{x}', y)\right\|_2} dx_1 dx_2 \tag{16}$$

Here $\beta$ is a constant and we obtain the key terms of the upper bound of adversarial transferability: $\left\|\nabla_{\tilde{x}} \mathcal{L}_{f_1}(\tilde{x}, y)\right\|_2$ and $\frac{\nabla_{\tilde{x}} \mathcal{L}_{f_1}(\tilde{x}, y) \cdot \nabla_{\tilde{x}'} \mathcal{L}_{f_2}(\tilde{x}', y)}{\left\|\nabla_{\tilde{x}} \mathcal{L}_{f_1}(\tilde{x}, y)\right\|_2 \left\|\nabla_{\tilde{x}'} \mathcal{L}_{f_2}(\tilde{x}', y)\right\|_2}$ in (16), which represent the gradient, i.e., loss field strength, and the relative relationship of two loss field lines, respectively.

However, performing a point-by-point integral operation is difficult due to computational problems. We can adopt a point estimate and take uniform points on the loss field line to generate unbiased regularization terms $\mathcal{R}$ based on the theoretical analysis above:

$$
\mathcal{R} = \lambda_1 \cdot \sum_{i=0}^{m} \sum_{j=0}^{m} \Theta(m,i,j) \left| \frac{\nabla_{\tilde{x}_i} \mathcal{L}_{f_1}(\tilde{x}_i, y) \cdot \nabla_{\tilde{x}_j} \mathcal{L}_{f_2}(\tilde{x}_j, y)}{\left\| \nabla_{\tilde{x}_i} \mathcal{L}_{f_1}(\tilde{x}_i, y) \right\|_2 \left\| \nabla_{\tilde{x}_j} \mathcal{L}_{f_2}(\tilde{x}_j, y) \right\|_2} \right|
$$
$$
+ \lambda_2 \cdot \sum_{i=0}^{m} \frac{1}{m+1} \left\| \nabla_{\tilde{x}_i} \mathcal{L}_{\mathcal{F}}(\tilde{x}_i, y) \right\|_2 \tag{17}
$$

$\Theta(m,i,j) = \frac{(m-i+1)(m-j+1)}{\left( \frac{(m+1)(m+2)}{2} \right)^2}$. $\tilde{x}_i$ is the $i$th adversarial examples of model $f_1$ and $\tilde{x}_j$ is the $j$th adversarial examples of model $f_2$. $m$ denotes the number of adversarial example points. $\lambda_1$ and $\lambda_2$ are hyper-parameters to adjust the weights of two terms. We constrain the two terms to promote the orthogonality and reduce the loss field strength of sub-models. Finally, we combine the two terms with the training loss on the original inputs to maintain the performance of benign data. Overall, the new loss function of training an ensemble $\mathcal{F} = \{f_1, f_2, ..., f_N\}$ including $N$ sub-models is (18):

$$
\mathcal{L}_{\text{new}} = \frac{1}{N} \sum_{i=1}^{N} \mathcal{L}_{CE}(f_i(x), y) + \frac{2}{N(N-1)} \sum_{i=1}^{N} \sum_{j=i+1}^{N} \mathcal{R}(f_i, f_j) \tag{18}
$$

We can also illustrate the optimization process of the second strategy of proactively improving the robustness of a deep ensemble as Figure 6. Although we analyzed the adversarial transferability within a deep ensemble from different perspectives compared with previous methods, some conclusions seem very similar. On the one hand, reducing the strength of the model's loss field can achieve the effect of enhancing the model's smoothness [75]. On the other hand, boosting the diversity of gradient orthogonality and reducing the magnitude of the gradient can also constrain the adversarial transferability [40,80], which can be explained as a special case of loss field orthogonality when the number of sampling points is $m = 0$.

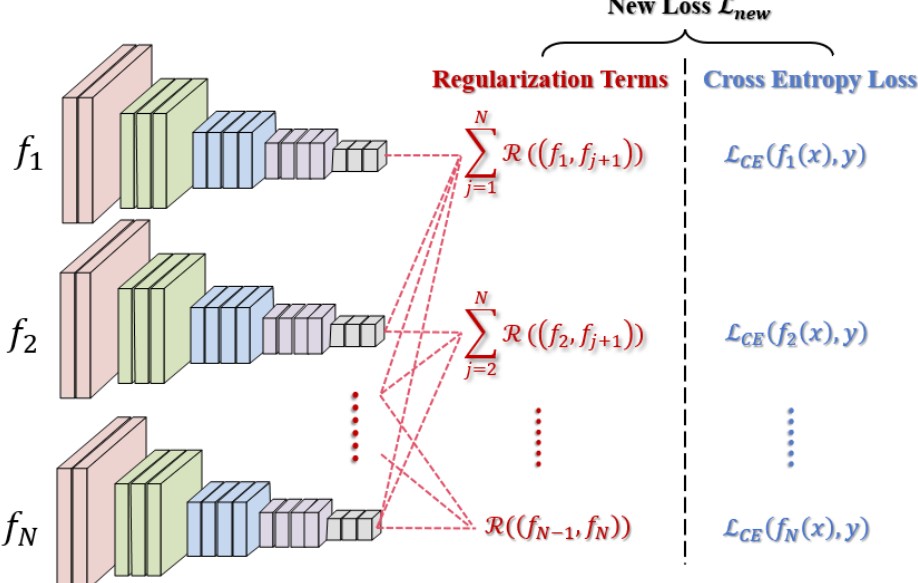

**Figure 6.** The illustration of optimization for the second strategy of proactively enhancing adversarial robustness of deep ensembles.

*3.3. Reactive-Proactive Ensemble Defense Framework*

We have introduced our strategies for deep ensembles in both proactive and reactive defense against adversarial RSIs, but they are still stand-alone components from each other. To form a linkage between them and further improve the adversarial robustness of the deep ensemble, we can build a reactive-proactive ensemble defense framework.

Under the ideal condition, a real-time RSI, whether it has been maliciously attacked or not, will be processed by the defense framework in an appropriate way, as illustrated in Figure 7. Once a working UAV with the defense system confronts the adversarial RSIs, the effective reactive defense can first reject them as TP with a high possibility. Even if some of them are wrongly detected as FN, the following proactive defense part will correctly recognize them because of high robust accuracy. In terms of normal scenarios, the reactive defense can pass the RSIs to the proactive defense, which will perform correct recognitions with high benign accuracy. If the detection rejects the RSIs, it will not maliciously influence the deep ensemble defense system.

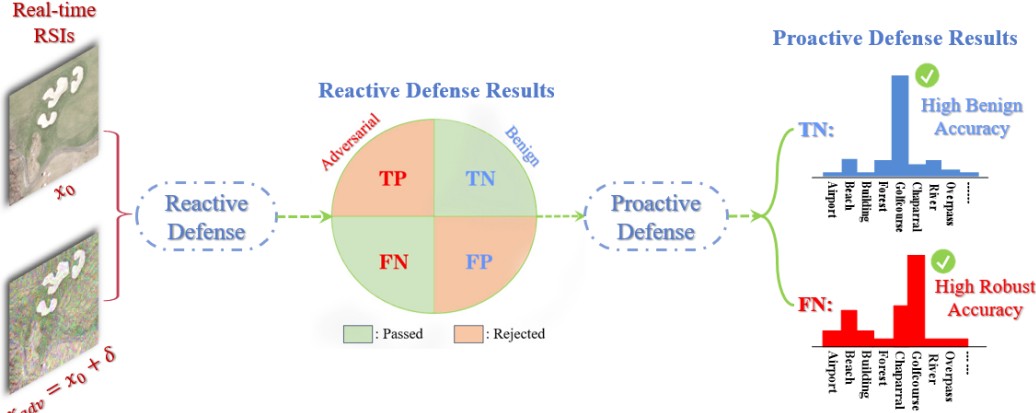

**Figure 7.** The desired results of processing the adversarial and benign RSIs with the proposed deep ensemble defense framework.

## 4. Experiments

*4.1. Datasets*

Three benchmark optical datasets used for RSI recognition are utilized in this article, including UCM (scenes), AID (scenes), and FGSC-23 (targets). The brief introductions of the three datasets will be presented in this section.

(1) UC Merced Land-use Dataset

The UCM contains 21 different scene categories, and each scene has 100 RSIs with a size of 256 × 256 and a spatial resolution of 0.3 m/pixel. The RSIs in this dataset are all selected from the U.S. Geological Survey (USGS) and captured from nationwide scenes across the country.

(2) Aerial Image Dataset

Different from the UCM for small scales in the whole dataset and each image, AID is a larger dataset with 10,000 RSIs and 30 different scene categories, each of which contains 200 to 420 RSIs with a size of 600 × 600. All the high-resolution images are extracted from Google Earth.

(3) Fine-grained Ship Collection-23

FGSC-23 collects 4052 high-resolution RSIs with 23 types of ships from Google Earth and GF-2 satellites for the recognition of ship targets. FGSC-23 is also characterized by diverse image scenes, fine classification, and complete labeling. We selected 17 out of the 23 categories because some categories do not contain square RSIs, which can greatly decrease the recognition accuracy.

### 4.2. Experimental Setup and Design

All the experiments are performed on the ensemble of three (i.e., $N = 3$) ResNet-18 with randomly initialized parameters. ResNet-18 is a relatively lightweight DNN architecture and is commonly used for resource-limited environments such as modern UAVs. Six popular adversarial attacks are used for the evaluations in each procedure, including FGSM, BIM, C&W, Deepfool, MIM, and PGD. BIM, MIM, and PGD are multi-step attacks whose attack iterations are all set to 20 with step size $\alpha = \epsilon/10$ where $\epsilon$ is the attack strength. For C&W attack, the number of attack iterations and constant $c$ are 1000 and 0.1, respectively.

We first train the deep ensembles as baselines with standard Ensemble Cross-Entropy (ECE) loss and 80% randomly selected RSIs for each dataset. The baseline ensembles are tested with the remaining 20% RSIs. During the training time, they are optimized for 100 epochs with Adam algorithm [81], learning rate as 0.01 and weight decay as $1 \times 10^{-4}$. The benign accuracy and robust accuracy for each attack algorithm can be recorded on each RSI dataset.

For reactive defense, we chose to use the baseline ensemble to maintain the features extracted from sub-models and verify the effectiveness of the ensemble method on RSIs. The scoring functions $d(\cdot)$ within the ensemble include $d_1(\cdot) =$ local intrinsic dimensionality (LID) [36], $d_2(\cdot) =$ kernel density estimation (KDE) [35] and $d_3(\cdot) =$ mahalanobis distance (MD) [37]. We compare the performances of the deep ensemble method with several stand-alone adversarial detection methods on the single DNN for each attack algorithm. The training data include adversarial RSIs (labeled as 1) and the corresponding benign RSIs (labeled as 0). The results of evaluation metrics AUROC, AUPR, and DR are collected on the test sets of three RSI datasets and six attack algorithms (i.e., 18 scenarios in total).

In proactive defense, for the first strategy, we use PGD-AT and TRADES with the FSR module to enhance the robustness of each sub-model equally. The FSR module is attached between layer 4 and the final linear layer. The hyper-parameters $\tau$ in (8) is set to 0.01 and $\lambda_{sep} = \lambda_{rec} = 1$ in (12). The adversarial RSIs added to the training sets are generated from PGD-10 with attack strength = 8/255 and step size = 2/255. For the second strategy, we still use PGD to imitate loss field lines with $n = 2$ sampling points, and the attack strength is set to 0.02 with 10 iterations. Additionally, $\lambda_1 = 0.2$ and $\lambda_2 = 200$ in (18) are fixed for all RSI datasets. The benign accuracy, robust accuracy, and average latency time in both strategies are compared and analyzed for all the possible attack scenarios. The definition of white-box robust accuracy is the proportion of correct recognition of adversarial RSIs to all tested RSIs. All the hyper-parameters in the proactive defense have been fine-tuned with the best results among the attempts (e.g., $\tau$ is selected from 0.005, 0.01, 0.05, and 0.1, $\tau = 0.01$ with the best performance).

The above experiments are all based on the white-box scenario, where the adversaries have full access to the attacked DNN models. In addition, black-box robustness analysis is conducted by crafting adversarial RSIs based on baseline ensemble as the surrogate model to fool the target strengthened deep ensembles in both defenses. The black-box robustness is more important because the details of DNN models are difficult to acquire in the real world. To demonstrate the feasibility of the whole defense system, we extract the non-rejected parts (i.e., TN and FN) in the black-box attack scenario from the reactive defense and perform robust recognition with the deep ensembles to attempt to achieve the desired outcomes in Figure 6. All the experiments are implemented on two NVIDIA GeForce RTX 3090 Ti GPUs, Python 3.7 and Pytorch 1.8.

### 4.3. Experimental Results

#### 4.3.1. Adversarial Detection (Reactive Defense)

Figures 8–10 present the AUROC and AUPR for our reactive defense framework with deep ensembles and three stand-alone algorithms on the three RSI datasets. As we can see, the ensemble defense framework achieves the best result in 18 out of 18 scenarios (i.e., all scenarios). The ensemble method can yield values of more than 80% in all the

gradient-based attacks of FGSM, BIM, MIM and PGD, which are much better than those in Deepfool and C&W. Gradient-based attacks are visually more obvious in RSIs and as the quantitative results show, the single-step FGSM is the easiest to detect because BIM, MIM, and PGD iteratively calculate more subtle perturbations with less distance between the benign RSI. Deepfool finds the shortest path to go across a decision boundary, and the optimization-based C&W generates the smallest perturbations among these attacks. In both harder scenarios, the deep ensemble framework still shows better capability than individual detectors. For instance, we can find an obvious enhancement in the UCM dataset with the two attacks. LID only outputs the results of 57.91 and 65.22, while the values of our modified ENAD framework are increased to 75.73 and 82.29.

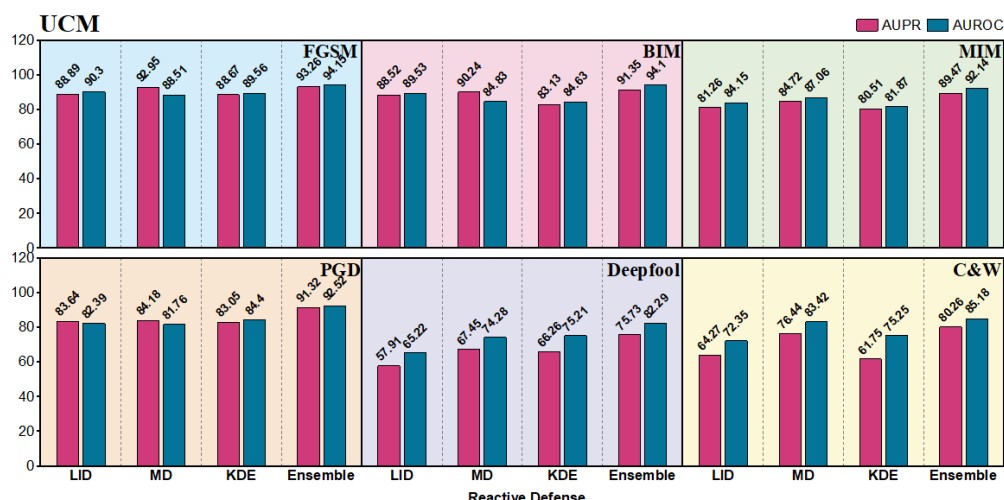

**Figure 8.** Results of our reactive defense framework based on deep ensemble and three stand-alone adversarial detectors for comparison on UCM scene recognition dataset.

The best result comes from using the ensemble method to detect FGSM on UCM with the AUROC and AUPR of 93.26 and 94.15. The general performances on AID are slightly lower than those on UCM, probably because the size of UCM is smaller, and the generated perturbations on UCM can be more concentrative for detection. In addition, some of the RSIs in FGSC are not square, which increases the difficulty and makes the detection results a bit lower than UCM as well. Overall, our deep ensemble model in reactive defense shows great potential against adversarial attacks in UAV-based scene classification.

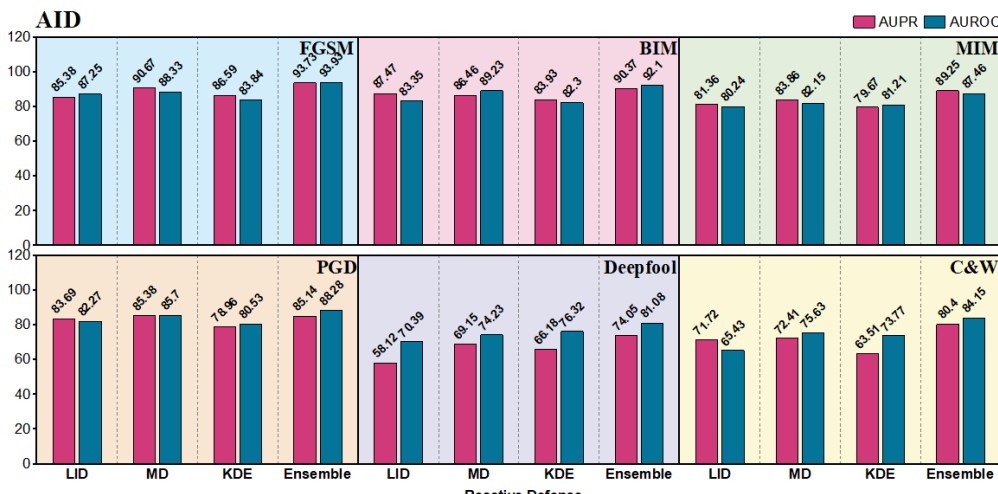

**Figure 9.** Results of our reactive defense framework based on deep ensemble and three stand-alone adversarial detectors for comparison on AID scene recognition dataset.

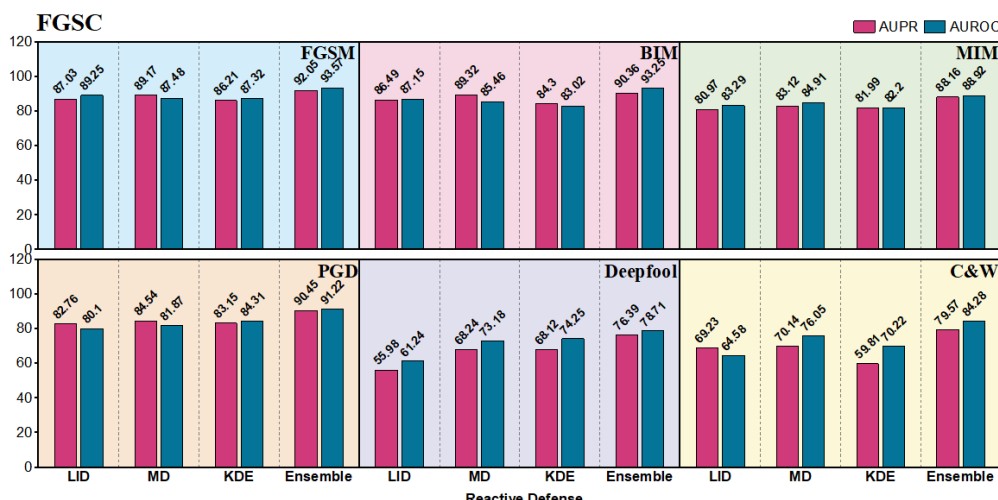

**Figure 10.** Results of our reactive defense framework based on deep ensemble and three stand-alone adversarial detectors for comparison on FGSC target recognition dataset.

### 4.3.2. Robust Recognition (Proactive Defense)

(1) Strategy I

We report the effectiveness of improving the robustness of each sub-model to strengthen the whole deep ensemble in Tables 1–3 (The best result in a row is highlighted in bold format). The white-box attacks ($\epsilon$ is the perturbation scale for gradient-based attacks) make the baseline ensemble severely damaged in recognition. The deep ensembles with sub-models trained by PGD-AT and TRADES can improve the robust accuracy under all the attacks for three RSI datasets. Especially for gradient-based attacks FGSM, BIM, MIM, and PGD, the increases are obvious. After applying the FSR module, the recognition performances are further improved in most attack scenarios. For example, BIM with larger perturbation makes the recognition accuracy only remain 4.76% in UCM. The results in PGD-AT and TRADES are increased to 24.76% and 24.05%, respectively. FSR modules attached to each sub-model further improve the robustness by 2.14% and 2.62% with a slight increase in computations. FSR modules can recalibrate the malicious non-robust activations and provide each sub-model with more beneficial features for recognition tasks.

However, for the benign RSIs, PGD-AT and TRADES only focus on the robust features during the training, leading to an obvious drop in benign data. The FSR module also has a negative effect on benign recognition performances because there are no malicious activations to be separated and recalibrated. For example, using the AID dataset, the benign accuracy is decreased by 1.45% and 2.75% for PGD-AT and TRADES, respectively. The mode of AT+FSR is not applicable to all scenarios. Deepfool can be an exception in that the enhancement of PGD-AT and TRADES are not pronounced, and adding FSR can even reduce robust accuracy. That is because the sub-models trained with gradient-based adversarial RSIs do not learn to recognize the malicious RSIs with that pattern. Deepfool makes the adversarial RSIs very close to the original ones. Therefore, the difference in the way that the adversarial RSIs are generated makes the malicious non-robust activations after FSR separation not involve fooling the models.

**Table 1.** White-box Robust Accuracy (%) for the Proactive Defense Method of Improving the Adversarial Robustness of Each Sub-Model in a Deep Ensemble on UCM ($\epsilon$: attack strength).

| | $\epsilon$ | **Baseline** | **PGD-AT** | **PGD-AT + FSR** | **TRADES** | **TRADES + FSR** |
|---|---|---|---|---|---|---|
| Benign Data | 0 | **96.19** | 86.90 | 85.48 | 88.81 | 87.86 |
| FGSM | 8/255 | 49.52 | 64.05 | 68.57 | 68.09 | **71.19** |
| | 16/255 | 30.48 | 50.71 | 52.62 | 51.19 | **53.81** |
| BIM | 8/255 | 11.67 | 47.14 | **53.81** | 49.76 | 51.43 |
| | 16/255 | 4.76 | 24.76 | **26.90** | 24.05 | 26.67 |
| MIM | 8/255 | 13.57 | 42.62 | **44.29** | 40.95 | 43.81 |
| | 16/255 | 3.10 | 18.09 | **19.05** | 16.90 | 18.81 |
| PGD | 8/255 | 10.95 | 51.43 | 54.05 | 53.57 | **56.67** |
| | 16/255 | 2.14 | 35.48 | 37.38 | 35.71 | **38.81** |
| Deepfool | – | 5.48 | 7.62 | 5.71 | **8.10** | 6.19 |
| C&W | – | 8.57 | 21.19 | 24.52 | 24.05 | **27.14** |

**Table 2.** White-box Robust Accuracy (%) for the Proactive Defense Method of Improving the Adversarial Robustness of Each Sub-Model in a Deep Ensemble on AID ($\epsilon$: attack strength).

| | $\epsilon$ | **Baseline** | **PGD-AT** | **PGD-AT + FSR** | **TRADES** | **TRADES + FSR** |
|---|---|---|---|---|---|---|
| Benign Data | 0 | **88.45** | 80.05 | 78.60 | 82.65 | 79.90 |
| FGSM | 0.05 | 39.25 | 56.40 | 59.95 | 58.55 | **61.25** |
| | 0.1 | 15.60 | 38.00 | 41.90 | 39.15 | **43.40** |
| BIM | 0.05 | 4.85 | 42.45 | **45.25** | 41.20 | 43.80 |
| | 0.1 | 0.90 | 16.65 | 18.55 | 17.40 | **20.00** |
| MIM | 0.05 | 7.60 | 42.50 | **43.35** | 40.15 | 43.05 |
| | 0.1 | 1.45 | 12.85 | **14.75** | 13.10 | 14.30 |
| PGD | 0.05 | 8.70 | 49.20 | 52.25 | 50.10 | **53.45** |
| | 0.1 | 1.35 | 20.25 | 21.80 | 22.35 | **23.52** |
| Deepfool | – | 7.20 | 7.25 | 4.80 | **8.70** | 4.90 |
| C&W | – | 8.85 | 18.65 | 20.50 | 20.05 | **22.85** |

(2) Strategy II

In terms of the second strategy, limiting the adversarial transferability among sub-models improves the robustness of the whole deep ensemble, we compare the robust accuracy of our proposed method based on the loss field with three advanced counterparts and the baseline. The three compared algorithms include ADP [72], DVERGE [40], TRS [75]. Tables 4–6 show the performances of robust accuracy for these deep ensembles against various attacks on three RSI datasets. As shown in the tables, all the algorithms can greatly improve the adversarial robustness of the deep ensemble and achieve much higher robust accuracy than that from the baseline ensemble. For instance, TRS can promote input gradient diversity and the model smoothness among the sub-models, which can enhance the recognition performance from 39.25% to 55.60% when the deep ensemble trained with AID dataset is targeted by weak FGSM attack.

**Table 3.** White-box Robust Accuracy (%) for the Proactive Defense Method of Improving the Adversarial Robustness of Each Sub-Model in a Deep Ensemble on FGSC ($\epsilon$: attack strength).

| | $\epsilon$ | **Baseline** | **PGD-AT** | **PGD-AT + FSR** | **TRADES** | **TRADES + FSR** |
|---|---|---|---|---|---|---|
| Benign Data | 0 | **84.10** | 73.22 | 70.29 | 74.06 | 71.55 |
| FGSM | 8/255 | 33.89 | 38.49 | 40.17 | 39.75 | **42.26** |
| | 16/255 | 20.08 | 25.52 | 26.36 | 27.61 | **28.87** |
| BIM | 8/255 | 12.13 | 23.01 | 24.69 | 25.94 | **28.03** |
| | 16/255 | 2.51 | 9.21 | **12.97** | 8.79 | 12.55 |
| MIM | 8/255 | 12.55 | 19.67 | **23.43** | 20.08 | 21.34 |
| | 16/255 | 1.67 | 7.11 | **11.71** | 6.69 | 10.04 |
| PGD | 8/255 | 6.69 | 29.29 | **32.21** | 28.45 | 31.38 |
| | 16/255 | 0.84 | 15.90 | **18.83** | 14.22 | 17.57 |
| Deepfool | – | 9.21 | 8.78 | 6.28 | **10.46** | 7.11 |
| C&W | – | 9.62 | 17.15 | 17.99 | 19.25 | **20.08** |

Our regularized deep ensemble for proactive defense can be the first choice among these strengthened ensembles because, in BIM, MIM, PGD, and C&W, the increases in robust accuracy are the highest. Especially in BIM and MIM with weaker perturbations, the superiority is the most obvious. However, compared with DVERGE, our ensemble is less robust against FGSM but much more robust against stronger multi-step attacks in BIM, MIM, and PGD. This may be because DVERGE pays attention to orthogonalizing the gradient of the original input, and therefore, they protect the gradient direction, while the proposed deep ensemble optimizes the whole loss field. For benign RSIs, all the algorithms maintain good performances, and ours are 85.95% on UCM, 83.25% on AID, and 80.33% on FGSC, which is slightly higher than those from other proactive defenses. Because the target recognition task is more difficult, the results of FGSC are generally lower than those of the other two RSI datasets. Overall, the second proactive defense strategy of limiting the adversarial transferability among sub-models also works well in enhancing the adversarial robustness of the whole deep ensembles. Our method of orthogonalizing the loss field and reducing the strength in the loss field can be a great attempt in the recognition task of the remote sensing field.

(3) Strategy I vs. Strategy II

Apart from comparing within one table, we also compare strategy I with strategy II by RSI datasets (i.e., Table 1 vs. Table 4, Table 2 vs. Table 5, Table 3 vs. Table 6) and analyze the applicable attack scenarios for both strategies if we only take the best result in each row.

In terms of FGSM attack, the best results of restricting the transferability within the deep ensemble are on par with those from enhancing each sub-model for both levels of perturbations on UCM and AID. In FGSC, strategy II performs obviously better than I for FGSM. When DNN-based UAVs suffer multi-step gradient-based attacks, such as BIM, MIM, and PGD, the analysis is not that simple. In BIM and MIM of scene recognition RSIs, the results of TRADES + FSR and PGD-AT + FSR are higher than those from strategy II when the perturbation strength is weak (i.e., $\epsilon = 8/255$). Our regularized deep ensemble based on the loss field performs much better with stronger perturbations (i.e., $\epsilon = 16/255$). For example, the highest value of BIM reaches 53.81% for weaker perturbations and 31.43% for stronger attacks. Moreover, strategy II yields more satisfactory performances in BIM and MIM attacks of target recognition (i.e., FGSC datasets). Because the models from strategy I are trained with PGD-attacked and benign RSIs, they are more effective against PGD than our regularized deep ensemble from quantitative results. Deepfool seems to be the most intractable attack, and the TRS algorithm from strategy II is slightly better than

any other one for all RSI datasets. Finally, as an optimization-based attack, C&W is also very imperceptible to human eyes. Our proposed algorithm in strategy II can be the most suitable method to solve C&W among all the attempts, but the highest value is only 36.25%. Therefore, there is still plenty of room for improvement in our future work.

**Table 4.** White-box Robust Accuracy (%) for the Proactive Defense Method of Limiting the Adversarial Transferability within a Deep Ensemble on UCM ($\epsilon$: attack strength).

|  | $\epsilon$ | **Baseline** | **ADP** | **DVERGE** | **TRS** | **Ours** |
|---|---|---|---|---|---|---|
| Benign Data | 0 | **96.19** | 85.24 | 82.62 | 82.86 | 85.95 |
| FGSM | 8/255 | 49.52 | 51.43 | **71.67** | 57.38 | 66.90 |
|  | 16/255 | 30.48 | 43.10 | **52.86** | 33.81 | 49.29 |
| BIM | 8/255 | 11.67 | 31.19 | 38.81 | 27.14 | **48.33** |
|  | 16/255 | 4.76 | 19.29 | 21.67 | 11.67 | **31.43** |
| MIM | 8/255 | 13.57 | 24.52 | 29.76 | 17.62 | **43.33** |
|  | 16/255 | 3.10 | 8.57 | 12.38 | 5.71 | **25.95** |
| PGD | 8/255 | 10.95 | 18.81 | 40.24 | 24.29 | **47.38** |
|  | 16/255 | 2.14 | 7.38 | 17.62 | 5.24 | **20.48** |
| Deepfool | – | 5.48 | 8.10 | 9.76 | **18.81** | 16.90 |
| C&W | – | 8.57 | 16.43 | **35.71** | 26.67 | 30.24 |

**Table 5.** White-box Robust Accuracy (%) for the Proactive Defense Method of Limiting the Adversarial Transferability within a Deep Ensemble on AID ($\epsilon$: attack strength).

|  | $\epsilon$ | **Baseline** | **ADP** | **DVERGE** | **TRS** | **Ours** |
|---|---|---|---|---|---|---|
| Benign Data | 0 | **88.45** | 77.90 | 76.15 | 82.70 | 83.25 |
| FGSM | 0.05 | 39.25 | 42.25 | **62.70** | 55.60 | 62.45 |
|  | 0.1 | 15.60 | 21.25 | 40.80 | 29.05 | **42.40** |
| BIM | 0.05 | 4.85 | 29.80 | 37.10 | 19.20 | **41.60** |
|  | 0.1 | 0.90 | 13.05 | 16.55 | 8.80 | **29.85** |
| MIM | 0.05 | 7.60 | 33.35 | 43.30 | 22.45 | **54.80** |
|  | 0.1 | 1.45 | 17.10 | 21.60 | 5.95 | **33.40** |
| PGD | 0.05 | 8.70 | 20.75 | 28.55 | 18.15 | **33.60** |
|  | 0.1 | 1.35 | 4.30 | 10.45 | 5.70 | **14.85** |
| Deepfool | – | 7.20 | 8.90 | 9.55 | **19.65** | 16.80 |
| C&W | – | 8.85 | 14.30 | 24.85 | 21.15 | **36.25** |

**Table 6.** White-box Robust Accuracy (%) for the Proactive Defense Method of Limiting the Adversarial Transferability within a Deep Ensemble on FGSC ($\epsilon$: attack strength).

|  | $\epsilon$ | **Baseline** | **ADP** | **DVERGE** | **TRS** | **Ours** |
|---|---|---|---|---|---|---|
| Benign Data | 0 | **84.10** | 77.82 | 76.99 | 79.08 | 80.33 |
| FGSM | 8/255 | 33.89 | 38.91 | **60.67** | 44.35 | 55.65 |
|  | 16/255 | 20.08 | 19.67 | 35.56 | 25.52 | **37.24** |
| BIM | 8/255 | 12.13 | 24.27 | 29.71 | 17.57 | **37.66** |
|  | 16/255 | 2.51 | 9.21 | 10.04 | 7.95 | **15.90** |
| MIM | 8/255 | 12.55 | 17.99 | 25.10 | 8.37 | **32.22** |
|  | 16/255 | 1.67 | 5.86 | 8.37 | 2.09 | **15.90** |
| PGD | 8/255 | 6.69 | 20.92 | 22.17 | 10.04 | **31.38** |
|  | 16/255 | 0.84 | 2.51 | 10.88 | 1.67 | **12.13** |
| Deepfool | – | 9.21 | 7.53 | 9.62 | **18.41** | 15.48 |
| C&W | – | 9.62 | 10.46 | 21.34 | 16.31 | **33.47** |

### 4.3.3. System Performances for Black-Box Scenarios

Even though our used methods can greatly improve the adversarial robustness of deep ensembles against adversarial RSIs for recognition tasks, all the experiments above are white-box and black-box transfer scenarios that are much closer to reality, where attackers cannot learn the internal information of our deployed ensemble models. To better simulate a real-life incident, adversaries generate the malicious RSIs against the baseline ensemble, and we activate the proposed reactive-proactive ensemble defense framework as the countermeasure. Here, we select scene recognition task (i.e., UCM and AID dataset) as a case study, and their test sets are exploited to generate adversarial RSIs based on four representative attacks respectively: FGSM (single-step gradient-based), PGD (multi-step gradient-based), Deepfool (based on decision boundary) and C&W (optimization-based). The attack parameters are the same as before, and a greater attack strength is used for FGSM and PGD. The deep ensembles used in the proactive defense are based on the recommendations of white-box scenarios in Tables 1, 2, 4 and 5, which is shown as Table 7.

**Table 7.** The Deep Ensembles Training Algorithms Used in the Proactive Defense for Different Attacks

| **Attack** | $\epsilon$ **(Attack Strength)** | **Proactive Defense Algorithm** |
|---|---|---|
| FGSM | 16/255(UCM); 0.1(AID) | Our new loss (18) |
| PGD | 16/255(UCM); 0.1(AID) | TRADES+FSR |
| Deepfool | – | TRS |
| C&W | – | Our new loss (18) |

According to the work mechanism of reactive-proactive ensemble defense framework in processing the RSIs, Figures 7, 11, and 12 present the experimental results for the black-box scenario. For example, the number of RSIs in the AID test set is 2000, so there are 4000 RSIs involved in each experiment. From Figure 11, we can learn that the DRs of the reactive defense ensemble method are all higher than 80%, which is acceptable for rejecting most adversarial RSIs and passing most benign RSIs in this part. In the black-box transfer attacks, detecting Deepfool and C&W is still a bit difficult because of the imperceptibility in adversarial RSIs. The detectors that learn the patterns of white-box gradient-based attacks during the training time also fit on the black-box transfer RSIs very well, which can reach 93.25% on FGSM and 88.85% on PGD for the metric of DRs.

The reactive defense accepts TN and FN and rejects TP and FP. The accepted RSIs then receive the class recognition in the following proactive defense. As shown in Figures 11 and 12, the number of correct recognitions is labeled before the total number in both parts of each transfer attack. The recognition of TNs corresponds to the benign accuracy of each type of proactive defense. For the parts of FNs, the black-box transfer results cannot be the same as the ones from the white-box scenarios. We can notice that FGSM and C&W with our regularized deep ensemble perform very well in black-box scenarios, which is due to the adversarial transferability being greatly restricted among the sub-models. When several models within the deep ensemble correctly recognize the RSIs, the rest cannot be far off the mark. It is difficult to simultaneously confuse all the sub-models with completely different trained weights. TRS has 113/263 and 49/106 correctly recognized for AID and UCM, respectively, which is also good if considering the generally low recognition accuracy in the difficult attack of Deepfool. Even if trained with PGD adversarial RSIs, TRADES +FSR seems to be unsuitable for the defense of black-box transfer PGD attacks. The number of corrections is 47 out of 115 for AID, which is kind of low for the adversarial defense in the real world. In summary, the proposed reactive-proactive defense framework with deep ensembles performs very well with our regularized loss under FGSM and C&W black-box transfer attacks but still has to be improved on PGD and Deepfool in the black-box scenarios.

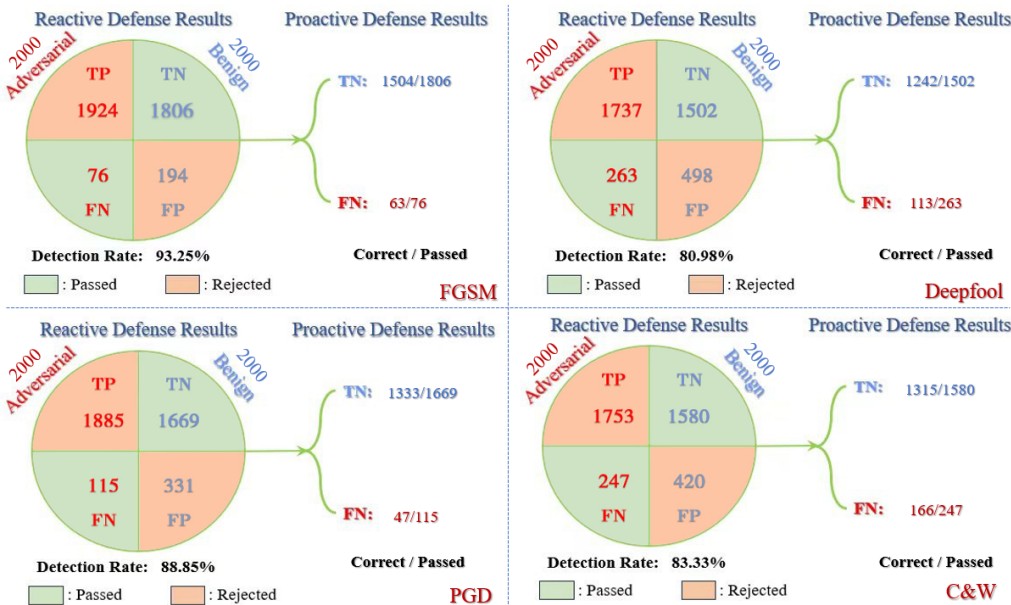

**Figure 11.** The results of black-box transfer attack against our reactive-proactive ensemble defense framework on AID where the proactive defense is based on the recommendations for different attacks in the white-box scenarios (the first quadrant: Deepfool; the second quadrant: FGSM; the third quadrant: PGD and the fourth quadrant: C&W).

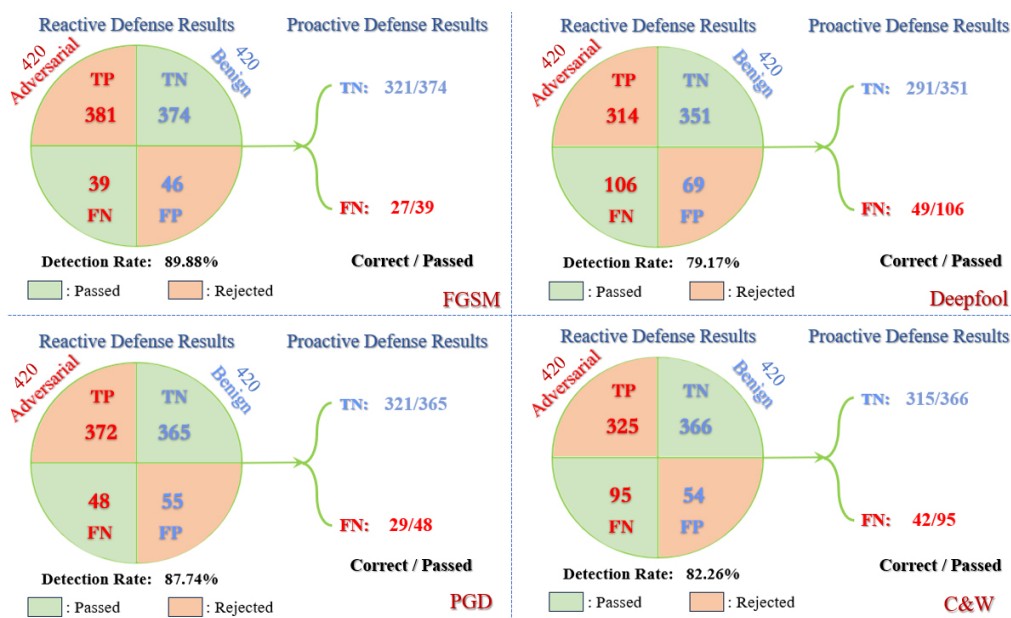

**Figure 12.** The results of black-box transfer attack against our reactive-proactive ensemble defense framework on UCM where the proactive defense is based on the recommendations for different attacks in the white-box scenarios (the first quadrant: Deepfool; the second quadrant: FGSM; the third quadrant: PGD and the fourth quadrant: C&W).

## 4.4. Discussion

In our experiments, we verified that deep ensembles with some designed modules or defensive training strategies can proactively alleviate the possible fraudulence in the adversarial RSIs for UAV-based intelligent recognition tasks. We attempt the FSR module to re-activate the non-robust channels during AT because most RSIs possess rich features and complex ground objects that can be implicitly beneficial to improve the recognition accuracy of the malicious RSIs. To demonstrate the activation of the non-robust channels after deploying FSR, we probe the activation frequency at layer 4 of one sub-model trained with PGD-AT + FSR on both AID and UCM dataset, which is visualized as Figure 13. The phenomenon of "complete rejection" in Figure 4 has been modestly boosted (i.e., the right region of both red and blue columns). This observation is still consistent for all classes. The activations for adversarial RSIs are not as frequent as for standard supervised training because of the existence of PGD-AT.

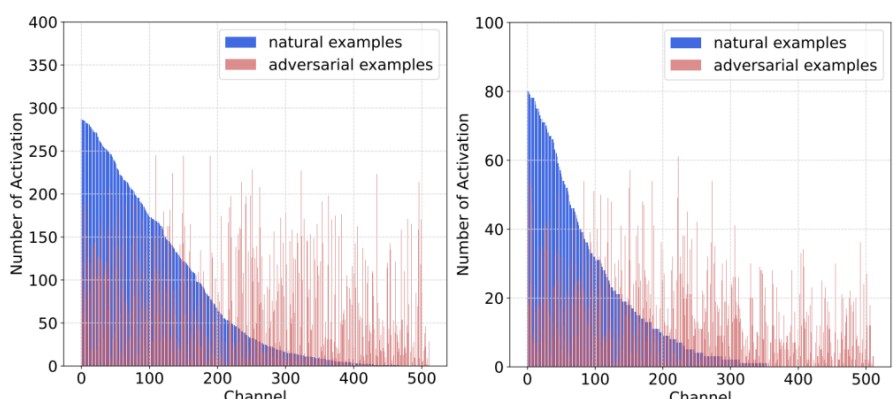

**Figure 13.** The activation frequency at layer 4 of ResNet-18 (512 channels) trained with PGD-AT + FSR (**Left**: AID dataset; **Right**: UCM dataset). Blue columns represent benign RSIs, and the red ones represent the PGD-20 adversarial RSIs.

For the strategy of limiting the adversarial transferability among sub-models in proactive defense, we visualize the decision boundary of ensembles and their sub-models in Figure 14 to better understand the adversarial robustness of the ensembles. Along the dashed line, we can find the reason why DVERGE is slightly more robust to single-step gradient-based attacks, as shown in Table 4.

Patterns of some colors (i.e., classes) in the other four methods can be very narrow along the negative direction of the gradient. Although the sub-models are sensitive to single-step attacks, our designed method shows higher robustness to multi-step gradient-based attacks for the whole ensemble. This proves that the enhanced robustness of our regularized deep ensemble may be largely derived from the reduction of adversarial transferability between sub-models. Additionally, the input (i.e., the central point) in ours is farther from the decision boundary than the others, indicating that this ensemble has the strongest defense against multi-step attacks among these methods, which corresponds to the results in Table 4.

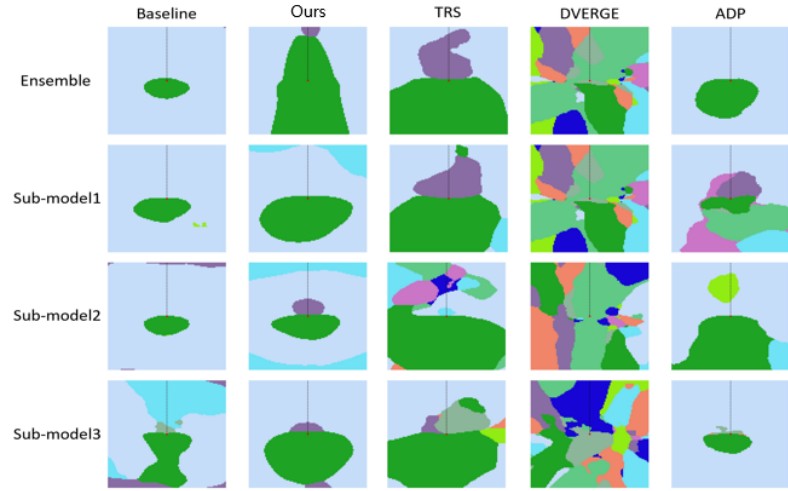

**Figure 14.** The decision boundary of ensembles and sub-models on UCM. The vertical axis is the negative direction of the gradient, while the horizontal is a random direction orthogonal to the gradient. The same color represents the same prediction, and each dashed line points in different directions.

More importantly, given that the applied scenario is RSI recognition on modern UAVs, the inference of the reactive defense method and all the enhanced deep ensembles need to be quick. We provide the average latencies of the reactive defense ensemble method and all proactive defense methods with one RSI in the UCM dataset in Table 8. All the inference time is less than 0.1 seconds. Even for the reactive-proactive defense framework, the time spent is about 0.1 seconds (modified ENAD + one of the proactive defenses). Therefore, all the deep ensembles are time-appropriate for real-world applications.

**Table 8.** The average latency of all the used ensemble methods with one RSI in UCM dataset

| Deep Ensemble Methods | Latency with One RSI (ms) |
| :---: | :---: |
| Modified ENAD | 85.71 |
| PGD-AT + FSR | 15.58 |
| TRADES + FSR | 17.63 |
| Our regularized ensemble | 15.16 |

## 5. Conclusions

Adversarial vulnerability is still an urgent and open issue in the remote sensing field. In this paper, we investigate the deep ensemble models for visual recognition tasks on

modern UAVs and enhance the adversarial robustness of the deep ensemble with the reactive-proactive ensemble defense framework for the first time. In the reactive defense, we modified the ENAD and achieved fusion on both feature and decision levels in one adversarial detection method. Most adversarial RSIs can be rejected at this stage, and the accepted ones are then passed to robust recognition. Different from other papers relevant to adversarial robustness enhancement, we attempted two mainstream strategies in proactive defense and analyzed their suitable attacks. The first is to enhance each sub-model's robustness, and we also introduce the FSR module to reactivate the suppressed non-robust features during AT. The second strategy is to restrict the adversarial transferability among the sub-models. From the perspective of the loss field, we promote the orthogonality of loss fields and decrease the loss field strength. We theoretically obtain two regularization terms for optimizing the whole deep ensemble model. From the experimental results on three RSI datasets, we verified that both strategies can greatly enhance the robustness of deep ensembles, and the second one can be better with multi-step attacks and large perturbation. To better adhere to the practical applications, we perform a case study with the whole framework in the black-box scenario. However, the paper also has some limitations. For example, there is more versions of adversarial training such as MART [82] and FAT [32]. We can test them with the FSR module in RSI recognition. Moreover, the attack strength for each RSI dataset is fixed, so there is a lack of plotting the accuracy vs. attack strength $\epsilon$ with different methods.

In the future, we will first make up for the limitations and develop more methods to enhance the robustness of deep ensembles. The deep ensemble trained with defensive strategies will also be deployed on physical UAVs and recognize real-time RSIs with adversarial perturbations.

**Author Contributions:** Methodology, Z.L.; software, Z.L.; validation, Z.L. and H.S.; original draft preparation, Z.L.; writing—review and editing, Z.L.; supervision, K.J. and G.K. All authors have read and agreed to the published version of the manuscript.

**Funding:** This work was supported by the National Natural Science Foundation of China under Grant 61971426.

**Institutional Review Board Statement:** Not applicable.

**Data Availability Statement:** The UCM, AID and FGSC dataset are all available in the references of this paper.

**Conflicts of Interest:** The authors declare no conflict of interest.

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
