# Peer review of "Adversarial Robust Aerial Image Recognition Based on Reactive-Proactive Defense Framework with Deep Ensembles"

_remotesensing, doi:10.3390/rs15194660_

Round 1
Reviewer 1 Report
This paper is well-written and this work is really important. I only have some minor issues:
- Please show some more ablation studies about how you tune the hyper-parameters in the network
- Could you add some discussions about the transferability of your method in this scenario? Such as you can refer to TGRS22-Partial domain adaptation for scene classification from remote sensing imagery
Reviewer 2 Report
Good work in adversarial robust aerial image recognition. Please solve the following comments with a minor revision and to be reviewed again.
1. Specify the dataset name used by this paper in Abstract.
2. The statement of contributions at the end of Sec. 1 should be enriched more (i.e., emphasize the value of combining the reactive defense and proactive defense, etc.), also aiming at better highlighting the characteristics of this paper.
3. Add more information about experimental platform.
4. Add the definition formula of white-box robust accuracy.
5. Sec. 3.3. Reactive-Proactive Ensemble Defense Framework is suggested to be placed in Sec. 3.1.
6. Add the definition of ϵ in Table 1.
7. In Table 1, which methods are designed for adversarial common object recognition and which methods are specifically designed for adversarial aerial image recognition?
8. IMHO, the Conclusion should be re-written to 1) explicitly describe the essential features/advantages of the paper that other paper do not have, and 2) describe the limitation(s) of the paper.
The English should be improved greatly.
Reviewer 3 Report
This paper proposed a reactive-proactive ensemble defense framework to solve the adversarial attacks on RSI datasets. The extensive experiments on three benchmark RSI datasets show that the proposed method could perform well on white and black-box attacks.
The paper is well-written and easy to follow, but I have some questions and suggestions as follows:
Methodology:
[1] For Eq (2), the a_l denotes the latent representation of the l th layer of the DNN, but what's the specific score function? Is the sigmoid?
[2]The dimension of each di is different; how to compose them together, and what's the size of the V?
[3] How do you choose the deep ensemble? Are they the same as the architecture pre-trained mode; or model trained with different epochs?
[4] What is the specific layer when applying the FSR to each sub-model?
Experiments:
[1] Could you provide the computational analysis of your ensemble methods?
[2] Please provide implementation details like learning rate and optimizer during the training.
[3] Could you add experiments with MART[1] and MART + FSR?
[4] Could you plot the accuracy vs. attack strength \epsilon with different methods?
References:
Recently, there has been significant progress in the development of adversarial attacks against deep learning models. Please add some references on defense for image [1,2,3], object detection [4,5], and EEG signal [6]. These references could help readers understand the adversarial attacks.
[1] Wang, Yisen, et al. "Improving adversarial robustness requires revisiting misclassified examples." International conference on learning representations. 2019.
[2] Qi-Zhi Cai, Curriculum adversarial training. In IJCAI, 2018.
[3] Yang, S., Guo, T., Wang, Y., & Xu, C. (2021). Adversarial Robustness through Disentangled Representations. Proceedings of the AAAI Conference on Artificial Intelligence, 35(4), 3145-3153.
[4] Yin, Mingjun, et al. "ADC: Adversarial attacks against object Detection that evade Context consistency checks." Proceedings of the IEEE/CVF Winter Conference on Applications of Computer Vision. 2022.
[5] Jia, Yunhan Jia, et al. "Fooling detection alone is not enough: Adversarial attack against multiple object tracking." International Conference on Learning Representations (ICLR'20). 2020.
[6] Li, Yunhuan, et al. "Adversarial Training for the Adversarial Robustness of EEG-Based Brain-Computer Interfaces." 2022 IEEE 32nd International Workshop on Machine Learning for Signal Processing (MLSP). IEEE, 2022.
Reviewer 4 Report
The experiments are well explained and the methodology is detailed. While the contributions are listed in the introduction, the objectives could also be mentioned in a concise manner. The results would be a bit more structured since there are so many figures to explain.
Extensive grammatical corrections are needed. I suggest having it thoroughly proofread.
Round 2
Reviewer 3 Report
The authors almost solved all my concerns, and I recommend accepting the paper.